25

2627

29

# Sea ice melt drives vertical pCO<sub>2</sub> variability modulating air-sea gas exchange

3 Henry C. Henson<sup>1,2</sup>, Dorte H. Søgaard<sup>2,3,7</sup>, Bjarne Jensen<sup>6</sup>, Kunuk Lennert<sup>4</sup>, Tim Papakyriakou<sup>5</sup>, Mikael K. Sejr<sup>1,2</sup>, Jakob Sievers<sup>6</sup>, Søren Rysgaard<sup>2,7</sup>, and Lise Lotte 5 Sørensen<sup>2,6</sup> 6 7 8 <sup>1</sup>Department of Ecoscience, Aarhus University, Aarhus, 8000, Denmark 9 <sup>2</sup>Arctic Research Center, Aarhus University, Aarhus, 8000, Denmark 10 <sup>3</sup>Greenland Climate Research Centre, Greenland Institute of Natural Resources, Nuuk, 3900, Greenland <sup>4</sup>UiT, The Arctic University of Norway, Tromsø, 9037, Norway 11 <sup>5</sup>Centre for Earth Observation Science, University of Manitoba, Winnipeg, MB, R3T 2N2, Canada 12 13 <sup>6</sup>Department of Environmental science, Aarhus University, Roskilde, 4000, Denmark <sup>7</sup>Department of Biology, Center for Ice-free Arctic Research, Aarhus University, Aarhus, 8000, Denmark 14 15 Corresponding author: Henry C. Henson (hch@ecos.au.dk) 16 17 18 19 20 21 22 **Key Points:** 23

- Spring melting of sea ice and snow introduces distinct heterogeneity in surface water conditions within coastal Arctic oceans.
- Standard bulk parameterizations for air-sea CO<sub>2</sub> flux calculations, based on subsurface pCO<sub>2</sub> measurements, may misrepresent flux direction and magnitude during melt periods.
- Vertical near-surface temperature and CO<sub>2</sub> gradients must be considered to improve flux estimates in stratified Arctic fjords.

https://doi.org/10.5194/egusphere-2025-5330 Preprint. Discussion started: 7 November 2025 © Author(s) 2025. CC BY 4.0 License.

#### Abstract

Strong spatial and temporal gradients in salinity, temperature, and carbonate chemistry in Arctic coastal surface waters complicate the estimation of air-sea CO<sub>2</sub> exchange, particularly during sea ice breakup. This study evaluates the applicability of the widely used bulk flux model under such conditions. The bulk approach assumes homogeneous surface conditions and linear vertical pCO<sub>2</sub> gradients. However, our observations in a stratified Arctic fjord reveal pronounced vertical variability in pCO<sub>2</sub> within the upper water column, including non-linear gradients near the air-sea interface. Micrometeorological measurements captured episodic upward CO<sub>2</sub> fluxes even when waters 1 m and below were CO<sub>2</sub>-undersaturated. We hypothesize that transient, high-pCO<sub>2</sub> layers at ~0.1 m depth intermittently decouple the atmospheric exchange from subsurface waters, reversing the expected flux direction. These findings highlight the importance of resolving near-surface variability during the transition from ice-covered to open water conditions. We recommend incorporating micrometeorological techniques and high-resolution vertical profiling in Arctic fjords to improve flux estimates of CO<sub>2</sub> in this rapidly changing region.

# Plain Language Summary

Sea ice melt adds less-saline water to the surface ocean. This creates vertical gradients in salinity, temperature, and partial pressures of carbon dioxide (pCO<sub>2</sub>). The concentration difference of pCO<sub>2</sub> across the air-ocean boundary is used to estimate gas transfer. Thus, the depth that we measure will impact our estimates. Directly measuring gas transfer showed CO<sub>2</sub> release from the ocean during sea ice breakup. This means ocean layering during ice melt may briefly reverse CO<sub>2</sub> transfer.

# 1 Introduction

High latitude coastal oceans are strong sinks for atmospheric carbon dioxide (CO<sub>2</sub>), absorbing more CO<sub>2</sub> per unit area than lower latitude regions (Dai et al., 2022; Roobaert et al., 2019). This strong uptake results from both the high solubility of gases in cold water and the intense biological activity typical of these regions. However, climate change is rapidly transforming this carbon sink. The Arctic is warming more than twice as fast as the global average, and sea-ice extent has been shrinking by over 13% per decade (Perovich et al., 2020). The loss of sea ice increases CO<sub>2</sub> uptake by exposing larger areas of open water for longer periods, which can further stimulate biological productivity (Arrigo and van Dijken, 2015; Bates and Mathis, 2009; Perovich et al., 2020). However, at the same time, melting sea ice freshens the surface layer and strengthens stratification, limiting vertical mixing with deeper water. Freshwater from melting sea ice and terrestrial run-off creates pronounced gradients in physical properties such as salinity and temperature, as well as chemical properties like dissolved inorganic carbon (DIC) and total alkalinity (TA) (e.g. Henson et al., 2025). As a result, the partial pressure of CO<sub>2</sub> (pCO<sub>2</sub>) can vary markedly with depth under melt conditions.

This vertical variability in pCO<sub>2</sub> poses a challenge for air-sea CO<sub>2</sub> flux estimation. The transfer of gases between the atmosphere and ocean depends on the difference in concentration between the two as well as the efficiency of the transfer process. Therefore, the bulk flux of CO<sub>2</sub> across the air-sea interface is commonly described as the product of the gas transfer velocity, k (m s<sup>-1</sup>), CO<sub>2</sub> solubility s (mol kg<sup>-1</sup> atm<sup>-1</sup>), and the partial pressure gradient ( $\mu$ atm) across the air-sea interface (Wanninkhof et al., 2009):

$$F = ks(pCO_{2_{sea}} - pCO_{2_{air}}) \tag{1}$$

 While widely applied, this formulation simplifies a complex process influenced by surfactants on the water surface, bubble-mediated gas exchange, and turbulence. Furthermore, surface water heterogeneity, driven by sea ice melt and freshwater runoff from land, complicate the physical and chemical processes governing air-sea CO<sub>2</sub> exchange. As a result, simplified parameterizations commonly used in global carbon flux estimates may be inadequate in these settings.

 In many studies, pCO<sub>2</sub> is measured several meters below the surface, assuming vertical homogeneity under well-mixed condition (Jørgensen et al., 2020). However, in stratified waters, where temperature, salinity, and pH can vary with depth, this assumption may lead to substantial errors in flux estimates (Ahmed et al., 2020; Dong et al., 2021; Miller et al., 2019; Watts et al., 2022). Although Arctic surface waters are often undersaturated with respect to atmospheric CO<sub>2</sub> levels and act as CO<sub>2</sub> sinks (e.g., Burgers et al., 2017; Dai et al., 2022; Henson et al., 2024; Laruelle et al., 2014; Roobaert et al., 2019), such assessments typically rely on sparse data collected from 0.5-5 m depth during limited periods. Dong et al. (2021) illustrate that high latitude fluxes of CO<sub>2</sub> calculated from the bulk method (based on measurements sampled at 6 m depth) differ significantly from those measured using direct eddy covariance.

Gas transfer velocity (k) is often parameterized as a function of wind speed. However, the true driver is mixing in the surface waters, which governs k. Fick's first law of diffusion, which underlies Equation (1), assumes a linear concentration gradient within the diffusive sublayer (Fig. 1) and steady-state conditions (Garbe et al., 2014). Jørgensen et al. (2020) argued that, due to seawater's high buffer capacity, chemical gradients do not significantly affect  $CO_2$  equilibration, supporting the use of measurements at 3-4 m depth. However, this conclusion relies on the assumption of horizontal and vertical homogeneity and neglects the effects of shallow surface stratification, particularly when alkalinity dilution is involved.

In Arctic spring, the upper ocean is often strongly stratified due to freshwater input from glacier melt, snowmelt, river runoff, and sea ice meltwater (Ahmed et al., 2020; Granskog et al., 2011; Meire et al., 2017; Miller et al., 2019). These inputs can extend vertical CO<sub>2</sub> gradients beyond the diffusive sublayer, complicating flux estimates during ice break-up and early open-water

periods. Several studies have demonstrated strong vertical heterogeneity in pCO<sub>2</sub> in Arctic coastal waters, with implications for air-sea flux calculations (Ahmed et al., 2020; Dong et al., 2021; Miller et al., 2019).

Surface freshening from ice melt and runoff strongly influences carbonate chemistry in Arctic coastal waters, which can either suppress or enhance oceanic CO<sub>2</sub> uptake. For example, Burgers et al. (2017) reported large horizontal variability in surface pCO<sub>2</sub> (144–364 μatm) linked to riverine input in the Eastern Canadian Arctic. Similarly, Sejr et al. (2011) observed strong surface pCO<sub>2</sub> gradients associated with salinity and temperature in Young Sound, and later documented a long-term decline in surface salinity (Sejr et al., 2017). Freshwater-induced stratification has also been shown to create vertical gradients in pCO<sub>2</sub> and pH with important implications for flux calculations (Miller et al., 2019). Finally, Bates et al. (2014) demonstrated that sea ice meltwater and melt ponds exhibit extreme variability in pCO<sub>2</sub> (<10 to >1500 μatm) and pH (6.1 to >10.8), highlighting the complex chemical landscape of ice-influenced waters. Together, these studies underscore the high spatial and temporal variability of carbonate chemistry in freshened waters across the Arctic.

To project future CO<sub>2</sub> uptake or outgassing in the Arctic, we must better understand the physical and chemical drivers of near-surface carbonate variability. In this study, we investigate the vertical and temporal variations in pCO<sub>2</sub> in a stratified Arctic fjord during sea ice breakup. By combining micrometeorological flux measurements with water-column pCO<sub>2</sub> profiles across the transition from ice-covered to open water, we evaluate the performance of the bulk flux model under Arctic seasonal transitions.

**Figure 1.** Schematic illustrating the interface between the air and the water in conjunction with pCO<sub>2</sub> concentration gradients. In equation 1, the concentration gradient is assumed to occur in the diffusive layer between the air and water, and the concentrations are assumed to be vertically constant in the turbulent layers. (Adapted from Wanninkhof et al. 2009)

#### 2. Study Site and Measurement Methods

# 2.1 Study Site

This study was conducted in Young Sound, a high Arctic fjord system located near the Daneborg Research Station in Northeast Greenland (Fig. 2). The fjord system comprises the Tyrolerfjord (inner fjord) and Young Sound (outer fjord), extending approximately 90 km from Tyroler River to the Greenland Sea. A sill at about 45 m depth separates Young Sound from the open ocean. Young Sound is 2 to 7 km wide, with an average depth of 100 m (maximum 350 m), and a total surface area of ~390 km<sup>2</sup>. Tidal amplitudes range from 0.8 to 1.5 m, with mean current velocities of approximately 2 cm s<sup>-1</sup> (Rysgaard et al., 2003). Freshwater inputs are primarily derived from Greenland Ice Sheet runoff, local glaciers, precipitation, and snowmelt from adjacent ice-free terrain. The drainage basin of the Tyrolerfjord/Young Sound system spans 2846 km<sup>2</sup>, of which 33% is glaciated.

Sampling was conducted from 12 to 31 July 2017. Sampling occurred during and immediately after a period of sea ice breakup. On 15 July, ice coverage was approximately 30%, decreasing to less than 10% by 16 July. Water sampling was conducted both from an inflatable boat and via sea ice leads, all in close proximity to the Greenland Ecosystem Monitoring (GEM) program's standard station (Fig. 2).

#### 2.2 pCO<sub>2</sub> Measurements Using the HydroC Sensor

Surface water pCO<sub>2</sub> was measured with a CONTROS® HydroC CO<sub>2</sub> sensor, which utilizes a membrane equilibrator coupled with a non-dispersive infrared detector. The instrument is equipped with a built-in water pump that provides flow rate of 35 ml s<sup>-1</sup> across the membrane. At each sampling depth, the sensor was allowed to equilibrate for 10 to 20 minutes, and values were recorded once stable for at least two minutes. The sensor operates over a range of 200-1000  $\mu$ atm and temperatures of -2 to 35°C. Annual calibration has been conducted using a certified 400  $\pm$  2% ppm CO<sub>2</sub> gas that was traceable to WMO standards. The sensor showed remarkable stability (397-401 ppm), supporting a measurement uncertainty of  $\pm$  2  $\mu$ atm.

#### 2.3 pCO<sub>2</sub> Estimation from TA and DIC

In addition to direct measurements, pCO<sub>2</sub> was calculated from total alkalinity (TA) and dissolved inorganic carbon (DIC) using the Seacarb package (Gattuso et al., 2024) in R. Due to the low salinity and cold temperatures characteristic of Arctic coastal waters, no universally accepted set of equilibrium constants (K1 and K2) exists. For consistency with previous studies in the region (Henson et al., 2023), we used the refitted constants from (Lueker et al., 2000). The selection of equilibrium constants introduces assumptions regarding seawater composition. (Raimondi et al., 2019) showed that different constants can lead to discrepancies between measured and calculated pCO<sub>2</sub> values, ranging from -3.1 to -35.8 μatm, with Lueker et al. (2000) demonstrating the best internal consistency under polar conditions. Still, (Sulpis et al., 2020) found that the calculation

of pCO<sub>2</sub> from DIC and TA can lead to uncertainty up to 15% under cold conditions, which is far greater than when pCO<sub>2</sub> is measured directly. 178

180

#### 2.4 Sea Ice TA and DIC Sampling

- TA and DIC in sea ice were assessed using three ice cores. Each core was sectioned into 5-10 cm 181
- segments and sealed in gas-tight NEN/PE bags with sampling valves (Hansen et al., 2000). 182
- Samples were transported in thermally insulated boxes to a nearby field laboratory. Cold (1°C) 183
- deionized water of known mass and carbonate composition (10 30 ml) was added to each bag, 184
- which was then resealed after removing air and weighted. 185

- The samples were melted in the dark over ~48 hours. Meltwater was transferred to 12 mL 187
- Exetainer vials (Labco, UK) pre-dosed with 20 µl of saturated HgCl<sub>2</sub> solution (5% w/v) to 188
- prevent microbial alteration. DIC was measured by on Apollo SciTech®'s AS-C3 analyzer while 189
- TA was determined via potentiometric titration on an Apollo SciTech AS-ALK2 total alkalinity 190
- titrator (Haraldsson et al., 1997). 191

193

#### 2.5 Physical Parameters

- Vertical profiles of conductivity, temperature, and depth (CTD) were obtained using a Seabird® 194
- SBE19plus CTD. On 16 July 2017, additional surface conductivity measurements were taken 195
- using a Thermo Orion-Star® instrument with an Orion 013610MD conductivity cell. Surface 196
- water temperatures were independently measured with a Testo® thermometer.

199

#### 2.6 Historical Data

For contextual comparison, pCO<sub>2</sub> time series data from the Greenland Ecosystem Monitoring program are also included in the analysis. pCO<sub>2</sub> data from 2007-2023 was measured using the same HydroC CO<sub>2</sub> sensor in August each year.

203 204

#### 2.7 Eddy Covariance

- Air-sea CO<sub>2</sub> fluxes, as well as sensible and latent heat fluxes, were estimated using
- micrometeorological instrumentation mounted on a 3-meter mast positioned approximately 0.5 206
- meters from the waterline. CO<sub>2</sub> concentrations were measured with a LI-COR® 7500 open-path 207
- gas analyzer, and three-dimensional wind vectors were recorded using a METEK® uSonic-208
- Scientific sonic anemometer. The authors recognize that open-path sensors over polar, marine 209
- environments can lead to larger errors due to the cross-sensitivity between humidity and CO<sub>2</sub> for 210
- near infrared gas analyzers. (Blomquist et al., 2014; Landwehr et al., 2014). However, the open-211
- path analyzer was used at the time due to its low power consumption, making it suitable for 212
- operation on battery systems in remote Arctic environments. 213

- To enhance reliability, we applied three complementary analysis techniques for flux estimation: 215
- (1) the standard eddy covariance (EC) method using EddyPro software (Version 7.0.6, LI-COR

Inc., 2019); (2) the dissipation technique (DT) (Sørensen and Larsen, 2010); and (3) the ogive optimization method (OGM) (Sievers et al., 2015a). Among these, the OGM was deemed most robust due to its ability identify and filter out low-frequency noise, sensor dampening, and large-scale turbulent motions that can bias flux measurements. These issues often introduce large relative bias associated with flux measurement over surfaces characteristically exhibiting low CO<sub>2</sub> fluxes, such as marine surfaces (Sievers et al., 2015b). OGM's superior ability to isolate relevant turbulent scales and reduce contamination from mesoscale variability is based on the accumulation and modelling of each cospectra over each 20 min averaging period (Fig. S1 and S2). Uncertainty in CO<sub>2</sub> fluxes was estimated directly from the OGM procedure. The reported values correspond to the standard error associated with the fitted ogive tail and reflect random uncertainty in flux integration.

**Figure 2.** Map of Greenland and the sampling area at the coast of Young Sound in Northeast Greenland. The red circle indicates the location of the Eddy Covariance tower while the Marine sampling site (Standard Station in the Greenland Ecosystem Monitoring program) is indicated as a blue circle (74.310, -20.300). Three Copernicus Sentinel true-color images of the fjord on July 12, 22, and 31 illustrate the transition between sea ice cover and open water.

#### 3 Data and results

# 3.1 CO<sub>2</sub> and Heat Fluxes

Surface air-sea CO<sub>2</sub> fluxes were measured using micrometeorological techniques between July 16 and July 31, 2017 (Fig. 3a). However, only a limited number of flux estimates passed the quality control criteria defined by OGM. This method uses a Haar wavelet analysis to assess the continuity of high-frequency CO<sub>2</sub> and vertical wind velocity signals, rejecting fluxes when either variable fails to meet spectral continuity thresholds. In addition to the automated filtering, manual inspection of the cospectra was performed to evaluate fluxes that were soft-flagged by the Haar analysis. Only fluxes that passed both stages of evaluation were retained for further analysis and are shown in Fig. 3a (Fig. S1).

Twenty-minute flux averages ranged from -25 to 110 mmol m<sup>-2</sup> day<sup>-1</sup>, with both upward and downward fluxes observed. Positive values indicate net efflux of CO<sub>2</sub> from the ocean to the atmosphere, implying temporary oversaturation of surface waters with respect to atmospheric CO<sub>2</sub>. These elevated fluxes occurred during and shortly after the sea ice breakup period. This finding contrasts with prior studies in Young Sound, which have described the fjord as a net CO<sub>2</sub> sink throughout the year. However, historic estimates are based on pCO<sub>2</sub> measurements from the month of August and taken at 1 m depth; not from vertical pCO<sub>2</sub> profiles that capture salinity gradients. Similar episodic outgassing events have been documented in other Arctic coastal systems under variable sea ice conditions, though particularly during or following sea ice melt (Butterworth et al., 2025; Else et al., 2011; Miller et al., 2011; Papakyriakou and Miller, 2011; Prytherch and Yelland, 2021; Sievers et al., 2015c).

In addition to CO<sub>2</sub> fluxes, eddy covariance measurements of sensible and latent heat fluxes were also quantified during the same period and are presented in Fig. 3b and 3c. For all scalar quantities, negative values represent downward fluxes directed toward the ocean surface. These heat flux data provide important context for interpreting variability in CO<sub>2</sub> exchange, as they reflect changes in atmospheric forcing and surface stratification. Corresponding meteorological variables, including wind speed and air temperature, are shown in Fig. 3d-f.

The calculated flux uncertainties are shown in Fig. S3 and illustrate CO<sub>2</sub> uncertainties were typically below 5 mmol m<sup>-2</sup> d<sup>-1</sup>. These uncertainties were used to evaluate the robustness of positive efflux events and to interpret flux magnitudes relative to measurement precision. Low uncertainties during both high uptake and efflux events demonstrated a good signal to noise ratio and provide support for the existence of variable uptake and outgassing. The highest uncertainties that exceed 5 mmol m<sup>-2</sup> d<sup>-1</sup> corresponded to near-zero fluxes, where precise fluxestimation becomes more difficult.

3.2 Surface Water pCO<sub>2</sub>

Vertical profiles of surface water pCO<sub>2</sub> were measured using the CONTROS® HydroC CO<sub>2</sub>
sensor across three distinct periods in July 2017 (Fig. 5a-c). Each observational period
corresponded to different sea ice conditions: before, during and after sea ice breakup (Fig. 2).
These high-resolution profiles revealed substantial vertical variability within the upper 2 to 3
meters of the water column. Under ice-covered conditions, pCO<sub>2</sub> measurements were taken
through an open melt pond. At this time, elevated CO<sub>2</sub> concentrations were observed at the very
surface (0.1 m), followed by a sharp decrease to approximately 1 meter depth, coinciding with

the ice-water interface. Below this depth, pCO<sub>2</sub> increased again, though remained well below

atmospheric concentrations (Fig. 4a).

During the period of sea ice breakup, when ice coverage ranged from approximately 30% to 10%, the vertical distribution of pCO<sub>2</sub> exhibited a similar structure. Concentrations were highest near the surface, declined to a local minimum at 1 to 2 meters, and then stabilized below 3 meters (Fig. 4b). Following, the complete breakup of sea ice, pCO<sub>2</sub> showed a more gradual decrease from the surface down to about 3 meters, beneath which concentrations remained relatively constant (Fig. 4c). Across all three observational periods, a shallow surface layer approximately 5 m thick was identified, within which most of the pCO<sub>2</sub> variability occurred. Below this depth, pCO<sub>2</sub> remained relatively constant.

These vertical structures are consistent with strong physical stratification, likely driven by freshwater input from glacial melt and surface heating. Temperature and salinity profiles collected concurrently support the presence of sharp vertical gradients in the upper water column, with salinity ranging from 1.4 to 29.6 PSU and temperature from -0.4°C to 6.2°C. These physical profiles, shown in Fig. 5, confirm that vertical mixing was strongly suppressed during the observational period.

Measurements from a different fjord in East Greenland on June 4, 2025, revealed strikingly similar vertical pCO<sub>2</sub> heterogeneity (Fig. 6). Elevated pCO<sub>2</sub> at 0.1 m decreased to a minimum around 1-1.5 m before increasing again and stabilizing near 3 m depth. Extreme stratification in the upper few meters caused pCO<sub>2</sub> levels in each profile to vary by more than 100 μatm between the surface and 1 m. This repeated observation of comparable vertical pCO<sub>2</sub> heterogeneity 8 years later and in a different fjord system suggest this is not an isolated phenomenon. Indeed, Arctic surface stratification induces chemical changes that may influence the way we estimate air-sea exchange of CO<sub>2</sub>.

**Figure 3.** The 5 min averages of measured fluxes and meteorological conditions over Young Sound during July 2017. This time period reflects the transition between sea ice break up (30% ice cover) and open water (no sea ice present) from 16 July 2017 to 1 August 2017. Air-sea exchange of (a) CO<sub>2</sub> (b) sensible heat and (c) latent heat were estimated using the ogive optimization method with estimated uncertainty shown as vertical error bars. (d) Wind speed, (e) wind direction and (f) air temperature are shown for the same period.

**Figure 4.** Measured Young Sound  $pCO_2$  profiles (a) prior to sea ice breakup (measured through open melt pond), (b) during sea ice breakup and (c) after sea ice break up measured through  $CO_2$  equilibration and calculation from carbonate chemistry parameters (DIC & TA).

329330

332333

**Figure 5.** Measured Young Sound profiles of under-ice water and open water salinity and temperature.

**Figure 6.** Measured  $pCO_2$  (a) and salinity (b) profiles at 4 locations in Tasiilaq Bay. Profiles were measured on June 4, 2025 following the method in Sejr et al. (2011).

**Figure 7.** Measured Young Sound pCO<sub>2</sub> profile after ice break up in 2017 compared with historical variation in pCO<sub>2</sub> at 1 m depth in the same location.

# 4. Discussion

Air-sea CO<sub>2</sub> fluxes in Arctic coastal areas are generally estimated using bulk parameterization models (Henson et al., 2024; Meire et al., 2015; Roobaert et al., 2019; Sejr et al., 2011). These models rely on several key assumptions, including unstratified surface conditions, a linear pCO<sub>2</sub> gradient within the diffusive boundary layer, and a vertically uniform pCO<sub>2</sub> profile within the mixed layer. Our observations challenge the applicability of these assumptions in Arctic coastal waters in several important ways.

#### 4.1. Evaluating the Constant pCO<sub>2</sub> Assumption

Vertical pCO<sub>2</sub> profiles collected during July 2017 revealed pronounced non-linear behavior in the upper 3 to 5 meters of the water column (Fig. 4). This directly contradicts the assumption that the ΔpCO<sub>2</sub> accurately represents the difference between the atmosphere and the "well-mixed bulk fluid" below the diffusive layer (Wanninkhof et al., 2009). Under ice-covered conditions, the lowest pCO<sub>2</sub> values (~150 ppm) were consistently observed just beneath the sea ice, with concentrations increasing with depth and stabilizing around 5 m (Fig. 4a). During the ice breakup

stage, a similar pattern emerged, although the minimum pCO<sub>2</sub> was higher (~250 ppm).

More recent measurements from Tasiilaq Bay in June 2025 demonstrate very similar vertical pCO<sub>2</sub> profiles. Indeed, 4 high-resolution profiles with measurements every 0.25 m reveal the same C-shaped pCO<sub>2</sub> variation. Like in Young Sound, the most elevated pCO<sub>2</sub> levels were observed near the surface, and pCO<sub>2</sub> minimums occurred near 1-2 meters depth before increasing and becoming stable. This repeated observation in a different fjord system, but during the period

of sea-ice breakup indicates this vertical variability may be representative during stratified Arctic conditions.

Several interacting processes influence surface water chemistry during ice breakup. Low surface water pCO<sub>2</sub> values reflect the influence of low-salinity meltwater from snow and sea ice or glacial meltwater found in freshened Arctic waters (Geilfus et al., 2015; Henson et al., 2025). However, surface water chemistry during the ice breakup period is further complicated by processes such as ikaite (CaCO<sub>3</sub>·6H<sub>2</sub>O) dissolution (Miller et al., 2011; Rysgaard et al., 2013; Søgaard et al., 2013) and high under-ice primary production (Søgaard et al., 2021). Additionally, snowmelt, characterized by low pH and ionic strength (de Caritat et al., 2005), may further alter carbonate system dynamics in the upper water column.

Two mechanisms may explain the nonlinear C-shaped trend in pCO<sub>2</sub> observed in the top few meters. First, as demonstrated by Henson et al. (2025) mixing between glacial meltwater and seawater can result in non-conservative behavior in pCO<sub>2</sub>, even when DIC and TA behave conservatively. In such cases, initial freshwater dilution leads to dramatically reduced pCO<sub>2</sub>, but at very low salinities, the diminished buffering capacity can cause acidification to occur and pCO<sub>2</sub> to increase again. Although, Henson et al. focused on glacial meltwater, our results suggest similar processes could occur in systems influenced by sea ice and snowmelt.

Both glacial meltwater and sea ice have low DIC concentrations and act to dilute the inorganic carbon of the surface ocean (Fig. S4). However, changes in alkalinity can also impact the buffering capacity of the water mixture, leading to nonlinear effects. If the meltwater has a lower TA:DIC ratio than seawater, due to the absence of ikaite, acidification and a shift in carbonate equilibria at very low salinities could lead to higher pCO<sub>2</sub> values at the surface. During July 2017, Young Sound showed both diluted DIC and TA levels in upper few meters, suggesting pH change during sea-ice break up could occur more easily (Fig. S4). Indeed, calculated pH profiles indicated variable surface conditions between periods of sea ice cover and sea ice breakup (Fig. S5). In this very fresh surface layer, diminished pH may elevate pCO<sub>2</sub> relative to waters around 1 m depth, where freshwater-seawater mixing ratios are more moderate and seawater buffering leads to very low CO<sub>2</sub> concentrations.

A second, but less likely, explanation involves atmospheric equilibration of sea ice melt ponds before draining into open leads. The relatively elevated pCO<sub>2</sub> observed at ~0.1 m depth could reflect such partial equilibration. While chamber-based studies (e.g. Geilfus et al., 2012, 2015; Nomura et al., 2010; Semiletov et al., 2004) have demonstrated both uptake and efflux of CO<sub>2</sub> in melt ponds, equilibrium times between melt-pond water and atmosphere depend upon pond depth, wind speed, and carbonate chemistry. For example, a 0.1 m deep pond under low wind conditions (~2 m s<sup>-1</sup>) may reach atmospheric equilibrium in 1-4 days. However, in our case,

401 pCO<sub>2</sub> values calculated from TA and DIC in melt ponds did not indicate equilibrium with the atmosphere, making this explanation less likely than the freshwater mixing mechanism. 402 403 Across all conditions, pCO<sub>2</sub> values calculated from TA and DIC using R-Seacarb were 404 consistently lower than in situ measurements from the CONTROS sensor (Fig. 5b, c). This 405 discrepancy supports the conclusion that chemical equilibrium is not achieved in the upper 2-4 m 406 during the melt season in Young Sound. Standard carbonate system calculations rely on 407 equilibrium constants that represent a set of reversible chemical reactions, assuming equilibrium 408 has been reached for each (Emerson and Hedges, 2008). However, in a dynamic surface 409 environment influenced by stratification, meltwater input, biological activity, and variable air-sea 410 exchange, this assumption of a steady state is likely violated, resulting in mismatches between 411 412 measured and calculated pCO<sub>2</sub>. 413 In addition, the equilibrium constants used in these calculations are typically derived from 414 laboratory experiments under conditions not representative of Arctic coastal waters. Most 415 constants were determined at higher salinities and temperatures that those commonly observed 416 during the Arctic melt season. Yet, temperature strongly impacts these thermodynamic constants 417 418 (K<sub>1</sub> and K<sub>2</sub>) (Cai et al., 2020). Indeed Sulpis et al. (2020) demonstrated that at temperature below 8°C, the use of constants from Lueker et al. (2000) may lead to underestimation of pCO<sub>2</sub>, which 419 could partially explain the discrepancies observed here. 420 421 As melt progresses and sea ice recedes, riverine input and vertical mixing become more 422 influential. Yet even after ice breakup, surface waters often remain fresh, and the resulting low 423 salinities help maintain stratification. In August 2017, vertical structure remained pronounced, 424 with elevated pCO<sub>2</sub> at 0.1 m which stabilized below ~3 m. In other words, near-surface 425 conditions remained decoupled from deeper waters. This persistent shallow layer, characterized 426 by low salinity, higher temperature, and elevated pCO<sub>2</sub>, suppresses gas exchange with the colder, 427 more undersaturated water below, consistent with observations by Dong et al. (2021). In such 428 429 environments, bulk flux models that assume homogeneity and linear gradients are likely to yield 430 biased or inaccurate estimates. 431 To place these 2017 measurements in historical context, we examined long-term surface water pCO<sub>2</sub> data collected at 0.5-1 m depth by the Greenland Ecosystem Monitoring (GEM) program 432 between 2007 and 2023. These data, measured using consistent protocols, are presented in Fig. 7 433 alongside our open-water profiles. Over the 17-year record, August pCO<sub>2</sub> concentrations at ~1 m 434 depth had ranged from 220 to 408 uatm and had consistently remained below atmospheric levels. 435 This apparent stability has contributed to the perception of sustained CO2 uptake throughout the 436 summer season. 437

However, the high-resolution vertical profiles obtained during the 2017 field campaign challenge

this assumption. Elevated pCO<sub>2</sub> levels confined to the uppermost meter of the water column may

quindetected in standard monitoring approaches that rely on fixed-depth sampling. These results suggest that short-lived but significant episodes of CO<sub>2</sub> outgassing can occur during rapid environmental transitions such as sea ice breakup. Consequently, existing sampling protocols may underestimate surface variability and bias flux estimates, especially in stratified conditions where near-surface chemistry is decoupled from subsurface layers.

# 4.2. Evaluating Bulk Model Flux Estimates

To assess whether bulk models are suitable for estimating CO2 fluxes in an Arctic fjord influenced by sea ice and snow melt, we calculated fluxes using seawater pCO<sub>2</sub> measurements from multiple depths and two gas transfer velocity parameterizations. Specifically, we computed fluxes throughout July using pCO<sub>2</sub> measured at 0.1, 1, 2, and 4 m. To estimate the surface (interface) pCO<sub>2</sub> at 0 m, we adjusted the 1 m measurements for the thermal skin effect based on skin temperature derived from sensible heat flux data (Fig. 3b), following the parameterization of Smedman et al. (2007). Accounting for this skin layer correction is critical, as Woolf et al. (2016) demonstrated that neglecting the thermal skin and relying only on bulk sea surface temperature can introduce significant errors in flux estimates. 

The resulting calculations (Table 1) show that estimated CO<sub>2</sub> fluxes vary significantly depending on the depth of the pCO<sub>2</sub> measurement. Notably, fluxes derived from 0.1 m differ markedly from those based on deeper values. Since many studies rely on pCO<sub>2</sub> measured at a fixed depth (often at 1 m or at a ships seawater intake below 5 m), these results underscore the potential for misrepresentation of flux direction and magnitude due to vertical heterogeneity in surface water chemistry.

Measured fluxes from eddy covariance (Fig. 3a) also exhibited large temporal variability. While micrometeorological methods integrate fluxes over a horizontal footprint, bulk flux models rely on single point pCO<sub>2</sub> gradients between air and water. Consequently, under heterogeneous or partially ice-covered conditions, the two methods are unlikely to yield identical results. However, meaningful comparison is still possible. Notably, agreement between micrometeorological and bulk model estimates was only observed in late July, and only when pCO<sub>2</sub> was corrected for the skin temperature. This aligns with the conclusions of Woolf et al. (2016) and Ford et al. (2024), who suggest that using skin-adjusted pCO<sub>2</sub> may improve flux estimates. Accurate application of this correction would require either direct measurements of skin temperature when sampling for the bulk method or high-resolution modeling of heat fluxes.

However, this agreement between methods did not hold before or immediately after ice breakup. During these periods, micrometeorological methods indicated episodes of large upward fluxes (48 mmol m<sup>-2</sup> d<sup>-1</sup>), while bulk model estimates suggested downward fluxes (-14 mmol m<sup>-2</sup> d<sup>-1</sup>). Although both methods yielded comparable magnitudes for downward fluxes, only the micrometeorological approach captured large upward events that could not be explained by

surface water pCO<sub>2</sub> profiles alone.

https://doi.org/10.5194/egusphere-2025-5330 Preprint. Discussion started: 7 November 2025 © Author(s) 2025. CC BY 4.0 License.

Measurements from both Young Sound and Tasiilaq demonstrate that during sea-ice breakup, pCO<sub>2</sub> levels are most elevated at the surface. This may be linked to acidification of the most freshened 0.5 m and a shift in the marine carbonate system, or partial equilibration due to air-sea gas transfer. If the air-sea boundary layer warms, for instance, due to solar radiation, pCO<sub>2</sub> may rise rapidly leading to oversaturation relative to atmospheric concentrations. Indeed, when pCO<sub>2</sub> measurements on July 31 were corrected for skin temperature (to estimate pCO<sub>2</sub> at the boundary layer), they revealed a transition from undersaturation to oversaturation (Table 1). While we did not directly observe this oversaturation in the vertical profiles, this likely reflects the inability to sample at the very surface layer. Moreover, profile measurements represent only single points in a system characterized by strong spatial and temporal variability during this seasonal transition. Nevertheless, the occurrence of C-shaped pCO<sub>2</sub> profiles during sea-ice breakup may help explain the observed reversal of flux direction captured by the micrometeorological approach.

Another potential explanation for the elevated CO<sub>2</sub> efflux that cannot be fully discounted involves cross-sensitivity between water vapor and CO<sub>2</sub> in open-path NDIR sensors (Blomquist et al., 2014). Although the OGM processing method is designed to minimize humidity-induced artifacts, it cannot entirely remove them. However, several lines of evidence indicate that the positive CO<sub>2</sub> fluxes are not solely an artifact of water vapor. Elevated CO<sub>2</sub> concentrations occurred during some, but not all, periods of high relative humidity, and extended intervals with similarly high humidity exhibited low CO<sub>2</sub> levels (Fig. S6). Moreover, positive CO<sub>2</sub> fluxes were observed even during negative latent heat flux events (Fig. S7a), which is inconsistent with a purely humidity-driven bias. In addition, CO<sub>2</sub> fluxes exhibited a strong negative relationship with sensible heat flux (Fig. S7b), consistent with physical air-sea exchange processes in marine environments. Taken together, these patterns support the interpretation that the observed effluxes represent real air-sea CO<sub>2</sub> exchange rather than being dominated by cross-sensitivity artifacts. Still, future studies using closed-path sensors during Arctic seasonal transitions are encouraged to verify the precise magnitude of CO<sub>2</sub> outgassing and uptake.

Overall, these findings echo those of Miller et al. (2019), who reported pronounced spatial heterogeneity in Arctic coastal pCO<sub>2</sub> and large differences in estimated flux depending on the sampling depth. The broader implications of this heterogeneity for seasonal or regional flux estimates remain unclear. However, if fluxes are upscaled from sparse, single-point measurements (e.g., once per month, as in Laruelle et al., 2014), substantial errors may result due to unrecognized spatial and temporal variability. Thus, our results emphasize the need for continuous, high-resolution observations of air-sea CO<sub>2</sub> fluxes, particularly in Arctic coastal systems affected by stratification and meltwater input. These observations are essential for refining flux parameterizations, reducing uncertainty in carbon budget estimates, and improving the representation of Arctic shelf systems in global carbon models.

**Table 1.** CO<sub>2</sub> fluxes calculated based on pCO<sub>2</sub> measured at the different depth. The fluxes are calculated using the bulk model of Ho et al., 2006 and Nightingale et al. (2000). We have used locally measured wind speeds for the calculations to match flux measurements captured by eddy covariance. The pCO<sub>2</sub> at 0 m is calculated from the pCO<sub>2</sub> measured at 1 m and then adjusted to the skin temperature, which is derived from the bulk equation and the measured heat fluxes.

| Date  | Depth<br>(m) | Temperature (°C) | Salinity<br>(psu) | pCO <sub>2</sub> (μatm) | Wind<br>Speed | Ho (2006) Flux (mmol CO <sub>2</sub> | Nightingale (2000)  Flux (mmol CO <sub>2</sub> | Eddy<br>Covariance<br>Flux<br>(mmol CO <sub>2</sub> |
|-------|--------------|------------------|-------------------|-------------------------|---------------|--------------------------------------|------------------------------------------------|-----------------------------------------------------|
|       |              |                  |                   |                         |               |                                      |                                                |                                                     |
| 6-Jul | 0.0          | 3.0              | 23                | 252                     | 6.8           | -14.88                               | -13.69                                         | 48.3                                                |
| 6-Jul | 0.1          | 3.0              | 23                | 244                     | 6.8           | -15.78                               | -14.52                                         |                                                     |
| 6-Jul | 1.0          | 1.8              | 26                | 240                     | 6.8           | -16.12                               | -14.83                                         |                                                     |
| 6-Jul | 2.0          | 1.1              | 28                | 241                     | 6.8           | -15.93                               | -14.66                                         |                                                     |
| 6-Jul | 4.0          | 0.3              | 29                | 275                     | 6.8           | -12.19                               | -11.21                                         |                                                     |
| 8-Jul | 0.0          | 6.0              | 7                 | 262                     | 3.3           | -3.45                                | -3.38                                          | -3.49                                               |
| 8-Jul | 0.1          | 4.3              | 7                 | 244                     | 3.3           | -3.98                                | -3.90                                          |                                                     |
| 8-Jul | 1.0          | 3.2              | 14                | 233                     | 3.3           | -4.20                                | -4.11                                          |                                                     |
| 8-Jul | 2.0          | 2.3              | 21                | 278                     | 3.3           | -2.86                                | -2.79                                          |                                                     |
| 8-Jul | 4.0          | 0.6              | 29                | 295                     | 3.3           | -2.34                                | -2.29                                          |                                                     |
| 8-Jul | 0.0          | 10.0             | 15                | 415                     | 2.5           | 0.53                                 | 0.54                                           |                                                     |
| 8-Jul | 0.1          | 10.0             | 15                | 405                     | 2.5           | 0.38                                 | 0.38                                           |                                                     |
| 8-Jul | 1.0          | 7.0              | 21                | 365                     | 2.5           | -0.22                                | -0.23                                          |                                                     |
| 8-Jul | 2.0          | 5.0              | 27                | 282                     | 2.5           | -1.43                                | -1.45                                          |                                                     |
| 8-Jul | 4.0          | 2.0              | 30                | 290                     | 2.5           | -1.32                                | -1.34                                          |                                                     |
| 1-Jul | 0.0          | 12.0             | 15                | 401                     | 2.0           | 0.20                                 | 0.21                                           | 5.07                                                |
| 1-Jul | 0.1          | 10.0             | 15                | 343                     | 2.0           | -0.36                                | -0.38                                          |                                                     |
| 1-Jul | 1.0          | 8.0              | 21                | 338                     | 2.0           | -0.40                                | -0.42                                          |                                                     |
| 1-Jul | 2.0          | 5.0              | 27                | 282                     | 2.0           | -0.92                                | -0.97                                          |                                                     |
| 1-Jul | 4.0          | 2.0              | 30                | 294                     | 2.0           | -0.81                                | -0.85                                          |                                                     |

530 **5 Conclusions** 531 During the summer thaw, carbon chemistry and pCO<sub>2</sub> dynamics in Arctic coastal surface waters 532 are significantly altered by the combined effects of snow and sea ice melt, terrestrial runoff, and 533 biological activity. These influences lead to substantial variability in surface temperature, pH, 534 dissolved inorganic carbon (DIC), and total alkalinity (TA), ultimately disrupting carbonate 535 system equilibrium in the upper water column. As a result, estimating air-sea CO<sub>2</sub> fluxes using 536 traditional bulk models becomes highly uncertain during this period. 537 538 539 The sea ice breakup period, typically lasting 2-4 weeks, represents a particularly dynamic and complex phase in the annual cycle. Despite its brevity, this phase may have a disproportionate 540 541 influence on total summer CO2 uptake, given that open-water conditions in high Arctic fjords are limited to only 80-120 days per year (Sejr et al., 2011). 542 543 Improved flux estimates will require more detailed and spatially resolved investigations aimed at 544 developing and validating gas exchange parameterizations tailored to the highly stratified and 545 ice-affected conditions of Arctic fjords. In particular, new approaches are needed to estimate gas 546 547 transfer velocities over waters influenced by snow and sea ice melt. Exchange rates depend not only on the pCO<sub>2</sub> gradient between the atmosphere and surface water, but also on rapid, 548 nonlinear changes in surface water chemistry driven by the composition and volume of 549 550 meltwater and runoff. 551 Once more suitable parameterizations for gas transfer velocity are established, accurate flux 552 estimation will also require knowledge of the depth at which surface water pCO<sub>2</sub> becomes 553 vertically homogeneous. Profiling pCO<sub>2</sub> in the upper water column is therefore essential to 554 555 identify this depth and to constrain surface flux estimates reliably. 556 Our study revealed large upward CO<sub>2</sub> fluxes using the eddy covariance (OGM) method, 557 particularly during the ice breakup period – fluxes that were not captured by the bulk model. 558 These events underscore the need for studies that integrate continuous, direct CO<sub>2</sub> flux 559 measurements with detailed observations of surface water carbonate chemistry, atmospheric 560 forcing, skin temperature, and turbulence at the air-ice-water interface. 561 562 Such integrated measurements are critical to improving our understanding of the frequency, 563 drivers, and net effect of upward CO2 fluxes in Arctic coastal systems. Ultimately, this 564 knowledge is essential to accurately quantify the seasonal and regional uptake of atmospheric 565 CO<sub>2</sub> in the rapidly changing Arctic. 566

570 Acknowledgments This study is a contribution to the GreenFeedback project (Greenhouse gas fluxes and earth 571 system feedbacks, Grant agreement: 101056921), funded by the European Union under the 572 Horizon Europe program, who also supported L.L.S and H.C.H's involvement. H.C.H. was 573 additionally funded by the AUFF (Aarhus Universitets Forskningsfond, project no. AUFF-F-574 2021-7-7) as part of his PhD. S.R. was funded by Aage V Jensens Fonde (grant no. AVJF21-575 3012) and the Danish National Research Foundation (grant no. DNRF 185). MKS was funded by 576 the POMP project (Horizon Europe grant: 101136875) and the Connecting the Dots project 577 (Villum Foundation grant: 50110) D.H.S received financial support from the Greenland Climate 578 579 Research Centre (GCRC), Greenland Institute of Natural Resources. The study also received financial support from The Danish Ministry of Climate, Energy and Utilities, Programme for 580 Arctic Climate, (project: Drivhusgas-observationer i Arktis (ObsArktis), 2017). Furthermore we 581 received support from The Arctic Research Centre, Aarhus University and Greenland Institute of 582 Natural Science. The authors especially wish to thank Egon Randa Frandsen, who assisted with 583 the logistics and the additional measurements in Young Sound. Additionally, the authors would 584 like to recognize the students in the EnCHil Nordic master program, who participated in taking 585 the Tasiilaq measurements. This work is a contribution to the Arctic Science Partnership (ASP) 586 and the MarinBasis component of the Greenland Ecosystem Monitoring Program. 587 588 **Author Contribution** 589 590 Conceptualization: LLS. Formal analysis, writing – original draft preparation: HCH. Funding acquisition: LLS, SR, MKS, TP. Investigation: DS, BJ, KL, TP, MKS, JS, SR, LLS. Writing -591 review and editing: DS, TP, MKS, SR, LLS. All the authors have read and agreed to the 592 published version of the paper. 593 594 595 **Data Availability Statement** Vertical profiles from both Greenlandic fjords can be found in the Zenodo data repository: 596 https://doi.org/10.5281/zenodo.17471918 597 598 599 **Competing interests** The authors declare no competing interests. 600 601 602 603 604 605 606 607 608 609

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
