# Peer review of "Sea ice melt drives vertical pCO2 variability modulating air-sea gas exchange"

_EGUsphere, 2025_

## Referee Comment (RC1)

The authors report surface-layer profiles of $pCO_2$, salinity, and temperature, which reveal pronounced gradients in these variables. They also present direct air-sea $CO_2$ flux measurements using an open-path eddy covariance (EC) system. I have many experiences with both open-path and closed-path EC systems. As also noted by the first reviewer, it is well established in the air-sea EC $CO_2$ flux community that open-path systems are sensitive to humidity cross-sensitivity (Landwehr et al., 2014; Blomquist et al., 2014; Nilsson et al., 2018).

The results presented here further suggest this issue: the strong correlation between the EC $CO_2$ flux and the heat flux (Fig. S7) is a typical indicator of cross-sensitivity effects. In addition, the back-calculation approach used by the first reviewer to infer an unrealistic $pCO_2$ value is, in my view, persuasive. Consequently, I do not have sufficient confidence in the EC $CO_2$ flux data presented in this study.

On the other hand, I find the profile measurements themselves to be very interesting and valuable. The data clearly demonstrate $pCO_2$, temperature, and salinity gradients from ~0 meter to ~10 meters in sea-ice melt regions, and notably show distinct vertical structures during different ice-melting periods. Even if the open-path EC $CO_2$ flux results were excluded, a manuscript focusing on these profile observations alone could still make a meaningful contribution to the community.

**Minor comments:**

- **Line 32:** Please define $CO_2$ at first use.

- **Line 34:** The assumption of homogeneity implies no vertical gradients. I understand that you may be referring to a linear gradient within the waterside mass boundary layer; however, the current wording could be misinterpreted as implying a linear gradient from the surface to several meters depth. I suggest revising to: *"The bulk approach assumes homogeneous surface conditions and no vertical pCO₂ gradients in the bulk seawater."* The sentence in line 36 can be revised accordingly by removing the word "non-linear."

- **Lines 37–38:** This sentence is unclear. Do you mean waters *at* 1 m depth? Please clarify.

- **Line 67:** You may consider citing Miller et al. (2019) here (https://doi.org/10.1029/2018GL080099).

- **Line 79:** Please add an appropriate reference.

- **Line 85:** "Most" may be more appropriate than "many."

- **Line 94:** For greater rigor, this statement could be revised to: *"Dong et al. (2021) illustrate that high-latitude $CO_2$ fluxes calculated using the bulk method (based on measurements at 6 m depth) differ significantly from those measured using direct eddy covariance in sea-ice melt regions."*

- **Line 135:** This figure originates from Liss and Slater (1974, *Nature*). You may want to indicate that it is adapted from Liss and Slater (1974) and Wanninkhof et al. (2009).

- **Line 176:** Please place the left bracket before the year and remove the comma.

- **Lines 212–213:** I understand the motivation here, but I suggest emphasizing that ensuring the robustness of the measurement technique should be the priority. Butterworth et al. (2025), cited later, demonstrate the feasibility of long-term $CO_2$ flux observations using a tower-based closed-path EC system.

- **Lines 253–254:** A reference appears to be missing.

- **Line 268:** Figure S2 has not yet been introduced in the text.

- **Line 452:** the derived skin temperature should be shown in the main text since this is a core parameter for the flux correction. I personally don't think the cool skin effect could be a significant factor that can affect the bulk flux for the data shown here. Because the cool skin effect is typically less than 0.2 K (Donlon et al., 2002), which is much smaller compared to the temperature gradients shown in Fig. 5. And 0.2 K will result in ~3 uatm decrease in the $\Delta p$CO2 (Dong et al., 2024, Sci. Adv.), which is also much smaller than the $p$CO2 gradients.

---

## Referee Comment (RC2)

The manuscript entitled "Sea ice melt drives vertical pCO2 variability modulating air-sea gas exchange" presents a study on the physical and chemical stratification of Arctic fjord waters during sea ice melt. The study delivers an important and timely message: near-surface stratification during the transition period can generate strong vertical pCO2 heterogeneity within the upper meters, which can substantially bias bulk air–sea CO2 flux estimates when sampled at a single depth.

The high-resolution vertical profiles presented are chemically plausible and provide a compelling explanation for how freshwater dilution and reduced buffering capacity can elevate pCO2 in the surface layer. This finding alone makes the study a valuable contribution to Arctic carbon budgeting.

However, there are significant concerns regarding the quantitative robustness of the Eddy Covariance (EC) fluxes presented. As noted by the community and other reviewers, the reliance on an open-path CO2 analyzer in a marine environment introduces susceptibility to water-vapor cross-sensitivity. The strong correlation between CO2 flux and latent heat flux suggests that measurement artifacts may be influencing the reported efflux magnitudes. Despite this, the study offers critical insights into the limitations of bulk parameterizations.

Therefore, I recommend publication pending revision, provided the authors explicitly acknowledge the open-path EC limitations by reframing EC fluxes as qualitative support (not definitive quantification) and keeping the focus on the robust vertical $pCO_2$ stratification findings, as this manuscript will be a strong reference for high-latitude air–sea exchange.

**Specific Comments:**

Line 34: The statement "The bulk approach assumes homogeneous surface conditions" should be clarified. It implies no vertical gradients within the water column, but strictly speaking, bulk models assume a linear gradient across the diffusive sublayer. Please revise to clarify that the assumption is a lack of vertical gradients below the interface.

Lines 167~ & 404~ : The manuscript attributes the persistent offset between calculated (TA-DIC) and measured pCO2 to "disequilibrium." However, carbonate system calculations in cold, low-salinity water carry substantial uncertainty regarding equilibrium constants. Before claiming physical disequilibrium, please expand the sensitivity analyses to determine if this offset is a physical phenomenon or simply a methodological artifact inherent to the constants used.

Lines 206~ & 495~ : There is a valid concern that the reported upward CO2 fluxes are biased by water-vapor cross-sensitivity, a known issue with open-path NDIR sensors in marine environments. The strong correlation between CO2 fluxes and latent heat fluxes (Fig. S7) is a typical indicator of this artifact, and the flux magnitudes (up to 100 mmol m$^{-2}$ d$^{-1}$) are difficult to reconcile with the surface pCO2 profiles without invoking extreme, unmeasured surface gradients. I strongly suggest reframing the EC data as qualitative evidence supporting the potential for outgassing rather than as definitive quantitative constraints, while explicitly acknowledging in the text that open-path cross-sensitivity likely inflates the reported flux magnitude.

Lines 450~ : The authors apply a skin temperature correction to the bulk flux calculations. Since the cool skin effect is typically small (< 0.2 K) and may not significantly alter the bulk flux direction compared to the observed large chemical gradients, if possible, please provide the derived skin temperature values in the text. This is necessary to justify whether this correction is a primary driver of the flux reversal or if the salinity/chemical gradient remains the dominant factor.

Figure S7: This figure is critical for assessing the quality of the EC data. Please ensure it is referenced in the main text when discussing the reliability of the fluxes.